# Inspection of Enamel Removal Using Infrared Thermal Imaging and Machine Learning Techniques

**DOI:** 10.3390/s23083977

**Published:** 2023-04-14

**Authors:** Divya Tiwari, David Miller, Michael Farnsworth, Alexis Lambourne, Geraint W. Jewell, Ashutosh Tiwari

**Affiliations:** 1Department of Automatic Control and Systems Engineering, University of Sheffield, Sheffield S1 3JD, UK; 2Rolls-Royce, Derby DE24 9HY, UK; 3Department of Electronic and Electrical Engineering, University of Sheffield, Sheffield S1 3JD, UK

**Keywords:** process inspection, terminations, infrared thermal imaging, machine learning

## Abstract

Within aerospace and automotive manufacturing, the majority of quality assurance is through inspection or tests at various steps during manufacturing and assembly. Such tests do not tend to capture or make use of process data for in-process inspection and certification at the point of manufacture. Inspection of the product during manufacturing can potentially detect defects, thus allowing consistent product quality and reducing scrappage. However, a review of the literature has revealed a lack of any significant research in the area of inspection during the manufacturing of terminations. This work utilises infrared thermal imaging and machine learning techniques for inspection of the enamel removal process on Litz wire, typically used for aerospace and automotive applications. Infrared thermal imaging was utilised to inspect bundles of Litz wire containing those with and without enamel. The temperature profiles of the wires with or without enamel were recorded and then machine learning techniques were utilised for automated inspection of enamel removal. The feasibility of various classifier models for identifying the remaining enamel on a set of enamelled copper wires was evaluated. A comparison of the performance of classifier models in terms of classification accuracy is presented. The best model for enamel classification accuracy was the Gaussian Mixture Model with expectation maximisation; it achieved a training accuracy of 85% and enamel classification accuracy of 100% with the fastest evaluation time of 1.05 s. The support vector classification model achieved both the training and enamel classification accuracy of more than 82%; however, it suffered the drawback of a higher evaluation time of 134 s.

## 1. Introduction

In high-volume manufacturing for electrical machines such as for the automotive sector, advanced manufacturing techniques supported by digital manufacturing tools have been deployed. Some of these techniques are transferrable; however, there are still gaps in activities and processes that require high degrees of human skill and cognition; typically undertaken within high-value, electrical machine manufacture; as found within the aerospace industry. According to a recent industry survey [1], the process of terminations or wire endings for Litz wire can often involve significant manual activities such as enamel removal and soldering, resulting in variations in quality. While terminating the stator windings in a connector or terminal box for onward connection to a drive or supply is a key process and it determines the electrical and mechanical quality of the joint, it is challenging to achieve precise process control due to the number and complexity of the influencing factors. This is especially relevant for Litz wires, which comprise several individually insulated and twisted thin wires. Litz wires have become a preferred choice for aerospace and automotive applications due to the advantages they offer, such as the reduced skin effect and proximity effect [2]. A review of the literature and discussions with the industry revealed that up to 10% of Litz wires have failed connections, due to incomplete stripping of some strands inside the connector. The solution to this problem lies in process monitoring and inspection of activities involved in the manufacturing of terminations. A review of the literature [1] has revealed a lack of any significant research in this area. Nevertheless, recent advancements in the field of sensing technologies and machine learning (ML) offer the opportunity to provide real-time inspection during the manufacturing of terminations.

The process of making terminations (joining) involves connecting a bundle of enamelled copper wire, composed of multiple individual strands, to a cable shoe/lug or an end sleeve. However, joining Litz wires is challenging due to the requirement of enamel (wire insulation layer) removal from every individual thin strand, followed by making an electrical connection between them. Recently, Seefried et al. [3] have compared and reviewed the challenges in various joining techniques for insulated Litz wires. The joining can be divided into two main process steps. The first step is stripping, which involves the removal of enamel from each strand to enable electrical contact with other strands and other components. The stripping process can be carried out by various methods, e.g., thermal, mechanical, or chemical stripping method [4]. The second step is joining, where the stripped ends are joined electrically and mechanically for making a connection. There are several processes for joining/contacting, such as soldering, ultrasonic welding, laser welding, and resistance welding.

Although individual processes of stripping and joining increase the handling effort, production times, and costs, they are inevitable in Litz wires that are required to operate effectively at higher temperatures for aerospace applications. In recent years, technologies that combine both the process steps of stripping and joining are being investigated due to their cost-effectiveness and efficiency [4,5]. However, quality control in processes such as stripping and soldering can be achieved by process monitoring and inspection at various stages in the joining process. There have been studies utilising ML approaches for some of the joining processes [6,7] and other electric machine manufacturing processes [8], but very limited research was found on ML approaches for the inspection of enamel removal and soldering. This research work investigates the use of infrared thermal imaging for the inspection of enamel removal on a Litz wire. This is an important step to ensure that all individual strands have had their enamel removed before they can be soldered for making a connection. The temperature profiles of the wires with or without enamel were recorded using infrared thermal imaging. Data were obtained during the heating and cooling profiles of the wires and then machine learning techniques were utilised for automated inspection of enamel removal on Litz wire bundles. The feasibility of several classifier models for identifying the remaining enamel on a set of enamelled copper wires was evaluated.

## 2. Summary of Key Joining Techniques and Machine Learning Technologies for Process Inspection

A summary of the main joining/crimping processes such as thermal crimping, ultrasonic crimping, welding, and soldering is provided in this section. The state-of-the-art in- process inspection utilising machine learning techniques has also been discussed for every joining technique. It was observed that very limited methods are available for inspection of enamel removal processes in Litz wires. This highlights the importance of the proposed non-destructive method for the inspection of enamel removal.

### 2.1. Thermal or Hot Crimping

Hot or thermal crimping is a widely adopted technology for terminating insulated copper wires where a pair of electrodes heated at around 500 °C are pressed to form a joint by evaporating the insulation layer of the copper wires [4]. A typical hot crimped termination is shown in Figure 1. This process offers the advantage of a reduction in production time by combining the process of stripping the insulation and connecting the cable lug. However, using a pair of electrodes heated at high temperatures and pressing against metals reduces their lifetime. The extreme forces and temperature to which the electrodes are exposed do not allow them to go beyond 5000 cycles [9]. To inspect the quality of the joint, end-of-line inspection methods such as continuity tests, resistance tests, pull-out force, and optical inspection are typically used by manufacturers [5]. To achieve process monitoring of quality for a hot crimped termination, Fleischmann et al. have proposed an artificial-neural-network (ANN)-based system for classification and prediction, that determines wear on electrodes and predicts joint quality by analysing process parameters such as variations in energy consumption and temperature [10]. Due to challenges associated with the tool wear, a majority of research focused on varying electrode geometries and materials [11], e.g., Keuhl et al. proposed a method where inductive heating was utilised to strip the wires, and the crimping tools were made of a nonconductive technical ceramic [4]. The National Aeronautics and Space Administration published a report on ultrasonic measurement techniques for quantitative assessment of wire crimp connections [12].

### 2.2. Ultrasonic Crimping

The ultrasonic crimping process utilises ultrasonic oscillations to make a connection. The cable lug with the inserted Litz wire is placed between the sonotrode and the anvil, and ultrasonic oscillations are induced. The oscillation damping and related friction processes cause heating in the contact zone, leading to the burning of the insulation and cold welding of the metallic parts. Ultrasonic crimping offers the advantage of higher tool life as compared to thermal crimping, but the disadvantage of this technique is that cables below 1 mm in diameter tend to dislocate from their initial position due to vibrations [13]. Inspection is typically conducted at the end of line using continuity tests, resistance tests, pull-out force, and optical inspection. Mayr et al. [14] proposed ML models that can predict the quality of a joint based on process parameters or its visual appearance. This work tested different regression models such as support vector machines (SVM), random forest, and AdaBoost, for estimating the withdrawal force of a crimped connection based on input parameters. In [15], SVMs were used to classify the quality of connections based on visual features.

### 2.3. Welding

Welding technologies, both traditional and laser, require prior removal of insulation from the contact area [3]. During the resistance welding process, force and current are applied to the bundle of Litz wire to compact it into a solid that then gets welded to the contact terminal placed on the lower electrode [3]. Tungsten Inert Gas welding is another example of a welding process that requires strict standards such as ISO10042 [16] to ensure the desired quality standards. This process is dependent on process parameters such as arc current/voltage, laser power [17], and welding speed. The process itself can be automated or manual, but the monitoring during the process is dependent on human supervision. Process monitoring using machine vision technologies presents limitations, due to the lighting conditions and contrast created by the arc light that can obscure the regions surrounding the weld. Previous work [18] compared real-time process monitoring of Tungsten Inert Gas welding using: machine vision methods and deep neural network (DNN) to monitor and perform defect classification of the welding process. It was found that both methods achieved high levels of accuracy in tracking the defects. The DNN solution was more adaptable, whereas the traditional method was more efficient. Other approaches, e.g., by Mayr et al. presented a system architecture for analysing the welding quality of joints for hairpin windings. The process parameters from the welding machine and visual information from the cameras were utilised for developing a quality monitoring system using convolutional neural networks (CNN) [19]. Sumesh et al. predicted the welding quality based on acoustic signals during the process [20]. Two classification algorithms, J48, and random forest, were utilised for the classification between good welds and defects.

### 2.4. Soldering

The soldering of termination process requires prior removal of insulation from the Litz wire. According to the British Standards BS EN 50390-2004 [21], soldering of terminations is a process of joining metallic surfaces using solder without direct fusion of the base metals. Previous work confirmed that the quality of the soldered joint is highly dependent on the process parameters such as the temperature of the solder, immersion time in solder, and angle of immersion, making it dependent on human supervision for monitoring during the process. Currently, soldering is a highly manual process without much automation. Typically, soldered terminations for electrical wires are optically inspected for cracks and dimensions [21]. Although there is limited research in the inspection of the soldered joint in Litz wires, some research has been published on defect detection in printed circuit boards utilising deep learning algorithms [22], quality inspection of soldered joints in using CNN [23], a cascaded CNN architecture to locate the defect in a soldered region and then labelling it [24]. However, there is very limited research on automated inspection of the soldering process involved in the manufacturing of terminations.

### 2.5. Lack of Process Inspection Methods for Enamel Removal Processes

As discussed in Section 2, many joining techniques such as soldering and welding require prior removal of enamel from the Litz wires to enable them to make an electrical connection. As mentioned by Seefried et al. [3], the key challenge in contacting insulated Litz wires is the high number of single wires coated with thermally resistant primary insulation. In addition, the single wires must not be mechanically damaged during the enamel removal or joining process to maintain the conductive cross-section. A detailed discussion on enamel removal processes, along with their advantages and disadvantages, has been provided in [25]. One method is mechanical removal of enamel using rotating brushes, where the copper wire is inserted between two rotating steel brushes that remove the insulation that is then extracted by suction [25]. The insulation can be removed by thermal methods where the insulation is melted and burnt off at high temperatures. Laser-based enamel removal processes are also being used by the industry due to the high removal rates. As discussed in previous sections, discussions with the industry revealed that up to 10% of Litz wires have failed connections due to incomplete stripping of some strands inside the connector. Currently, the majority of the techniques used on the shop floor are destructive testing methods or post-analysis methods, such as computerised tomography scanning or the fluorescence measurement method [25]; currently, no non-destructive testing process for inspection of enamel removal has been published.

## 3. Materials and Methods

This section provides further information on the methods of the proposed solution and the experimental setup. It has been divided into two parts. First, a brief description of the use of the infrared thermal imaging technique for inspection of enamel removal on Litz wires is discussed. The second part describes the experimental setup; the process for recording data and training the machine learning models is also discussed.

### 3.1. Infrared Thermal Imaging for Inspection of Enamel Removal

Active infrared thermography (IRT) is an effective tool for non-destructively inspecting materials by irradiating the material with a heat source and analysing the resulting spread of heat for any irregularities. Several investigations have been performed into methods to heat the sample to inspect the defects. One of the methods is pulsed thermography (PT), where the target sample is heated with a pulse of thermal energy for instantaneous heating [26]. PT has been used to investigate the relationship between the coating thickness uniformity error and surface temperature change [27], and the relationship between phase angle and coating thickness [28]. Zhou et al. showed that PT can be used to inspect the damage inside composite materials [29]. Compared to other thermography techniques, PT offers the advantage of the availability of instruments to generate the heat pulse for heating the material under inspection, e.g., flash lamps offer excellent control over the power and duration of the heat pulse.

### 3.2. Experimental Setup and Process

An E6XT infrared camera from FLIR was used for experiments. It has a 43,200 (240 × 180) pixel infrared detector, −20 °C to 550 °C (−4 °F to 1022 °F) temperature range, and some internal memory for storing recordings. It needed to be connected to a computer for recording live data. The live data were recorded at a rate of 15 Hz and returned images of size 240 × 160. FLIR’s Atlas 6 Software Development Kit (SDK) was used to record live data from the IR camera. The SDK is written in C# and provides a series of tools for interacting with FLIR cameras and requesting data from them.

A pair of heat lamps (500 Watts) was used as a sustained heat source, and was set to achieve the required rise in surface temperature of the wires corresponding to a 2 KJ (nominal) surface energy exposure. The heating time of the lamp was varied between one and ten seconds to elicit a range of responses. The picture of the setup used for experimentation is shown in Figure 2.

The bundle of Litz wire was held in place on a Perspex sheet 4 mm thick using masking tape. The IR camera was held at a height of approximately 15 cm from the wires and was connected to a laptop via a USB cable. Flir Report Studio (“FLIR Thermal Studio Suite”) was used to trigger the recording of infrared radiation emitted from the wires. The data were obtained under different environmental temperatures, different heating times, varying distances from the wire, varying amounts of enamel on the wires, and different numbers of wires under inspection. The values were saved to a compressed sequence file (CSQ) containing temperature data for the recording as well as metadata about the device. The temperature data were exported as a Comma Separated Values (CSV) file and the image data as an Audio Video Interleave (AVI) video file. The image data were generated by false colouring the temperature data using FLIR’s Iron colour palette, where the limits to build the colormap were set at the start of the recording. To save file space, the CSV file was converted to a compressed Numpy NPZ file. A flowchart depicting the process of recording and training the models is presented in Figure 3. Python 3.7.4 was the primary development environment used to analyse the data and develop the models. The sklearn package was the primary source of machine learning models except for OpenCV 4 [30]. It was used for its suite of image processing tools and implementation of the Gaussian Mixture Model.

## 4. Results and Discussion

A bundle of Litz wire with enamel coating of polyamide-imide had some wires with enamel, some wires partially removed, and some with enamel fully removed. The wire ends were exposed to a heat lamp for 10 s and then cooled. The infrared images of the wires during the heating and cooling cycle were recorded. The images obtained by the IR camera during the heating and cooling cycles are shown in Figure 4. The bright areas in the images represent higher temperatures. The wire labelled as 1 in the picture had its enamel completely removed and, therefore, had copper on the surface, the wire labelled as 2 had a coating of enamel on the surface, and wire 3 had its enamel coating partially removed. It was observed that the thermal profile of the three wires was different during the cooling cycle; wire 1, which did not have the enamel coating, cooled faster than wires 2 and 3, which had enamel on their surface. This is due to the difference in emissivity of copper and enamel (polyamide-imide) as approx. 0.2 and 0.9, respectively.

After capturing the IR imaging data from the heating and cooling cycles, image processing techniques were applied to distinguish the wires/areas with and without enamel coating. The wires were successfully detected from the IR image using the Canny edge detection algorithm, and then the generated mask of detected edges was dilated to isolate the detected wires from the background image [31], as shown in Figure 5. The simple linear iterative clustering algorithm groups pixels based on their physical proximity and colour. This algorithm was investigated for segmenting the IR image into smaller regions (superpixels) [31]. The image processing algorithms performed well when the data were ideal and the enamel was visible; however, the performance was not consistent across the range of data collected.

### 4.1. Thermographic Signal Reconstruction (TSR)

A standard approach to processing IR imaging data is to apply thermographic signal reconstruction (TSR) whereby a log polynomial is fitted to the temperature history of each pixel in the thermogram [32]. The fitting reduces noise by not including it in the fitting and compresses the image, reducing file size and allowing faster analysis [33]. The result is an N-channel matrix of model coefficients. The log polynomial fitting was applied to the data obtained during the cooling cycle of the wires. As discussed in the previous section, due to the difference in cooling rates of copper and enamel, the cooling rate of the samples contains the information needed to differentiate between the different materials. Log and standard polynomial models of 5th, 6th, and 7th order were fitted to the collected data with the increasing order having little impact on the overall fitting. The plot in Figure 6 shows the TSR fitting to recorded data using the 5th order polynomial and log polynomial.

One of the limitations of TSR is that it has to be fitted to the entire cooling period and each pixel, making the fitting time potentially very high. Iterating over each pixel of the thermal history would require a high computational load and processing time, making this approach unsuitable for a live system where it is expected to take only a few seconds for it to process new information. In addition to this, understanding the results from TSR would require special expertise not necessarily present in the shop-floor environment where the technology could be deployed. A fast, effective classifier model that assigns numerically simple labels would be better suited for a live system. The next approach was to train a classification model on the temperature history of the data.

### 4.2. Classification Models

Due to the difference in thermal conductivities of copper and enamel, analysis of the data recorded during the cooling period could aid in inferring/identifying the material occupying the pixel. For modelling purposes, data in the form of a feature set summarising the behaviour of the target period were required. It was found through investigation that the minimum, maximum, and average temperature over the cooling period varied more distinctly between different materials than other metrics. These metrics were applied to the recorded temperature values, their first derivative, and second derivative. The derivatives were investigated as, over the course of the recordings, the minimum and maximum temperature of the environment increased with each subsequent recording. The use of derivatives made the technique more robust to environmental changes caused by repeated heating of the wires. To train the models and to evaluate the results afterwards, a series of masks was constructed for each material present in the recordings (copper, enamel, background, and fixing tape), as shown in Figure 7. Sometimes the tape holding the wires down was included in the recording frames, so a mask for the tape was made as its thermal response was distinctive. This is likely to be due to the tape’s reflective surface.

Three different classifier models: a k-means clustering algorithm, Gaussian Mixture Model with expectation maximisation (GMM-EM), and support vector machines (SVM) were trained in two different ways. The first was to identify the cooling periods by their respective material, as specified by the training masks shown earlier. A separate model was also trained to classify whether or not the pixel contained enamel. The two-class model consisted of classes as enamel and copper, whereas the four-class model would contain classes as copper, enamel, background, and fixing tape. A two-class model would be ideal for a live system as it would return a binary image which would be easier to process. Both models were tested to see which would perform better.

#### 4.2.1. K-Means Clustering Algorithm

The first model tested was the popular k-means clustering algorithm. It is a common method for separating data into k-groups. The groups or clusters are defined by their centre and are iteratively adjusted as more data are introduced [34]. This well-defined algorithm could be easily trained. The training accuracy in the following plots is on a scale of 0 to 1, with 1 representing 100%. The training accuracy varied between approximately 20% and 90% for four classes (copper, enamel, background, and tape), Figure 8a. It also varied with different files which suggest the sensitivity of the model. The accuracy became more consistent for 2 classes when the 2nd derivative was used, achieving an accuracy of around 90% for 5 out of the 7 files in the plot, as shown in Figure 8b.

#### 4.2.2. Gaussian Mixture Model with Expectation Maximisation (GMM-EM)

A Gaussian Mixture Model with expectation maximisation was tested next. As proposed by Stauffer and Grimson [35] this method provides a means of classifying objects from the background of an active image. The model describes the image as a series of weighted Gaussian distributions based on statistical features. New data are classified by closest distribution. OpenCV’s implementation of the Gaussian Mixture Model was used, which employs expectation maximisation (EM) to learn the mixture [36]. The algorithm attempts to estimate the parameters of the distribution by finding those that would maximise the likelihood of describing the data [37].

The training accuracy in the following plots is on a scale of 0 to 1, with 1 representing 100%. The training accuracy varies between 20% and 90% for the 4-class model, and from 40–90% for the 2-class model, as shown in Figure 9. Choosing to train with the 1st and 2nd derivatives led to higher accuracy in the majority of cases for both the 2-class and 4-class scenarios. For the four-class model, the GMM training accuracy generally varied greatly from one file to the other. The accuracy became consistently higher when the 1st derivative was used for the 2-class model.

#### 4.2.3. Support Vector Machines (SVM)

Support vector machines (SVM) seek to find the hyperplane that separates multidimensional data into clusters [38]. Three different implementations were tested: C-support vector classification (SVC), Nu-support Vector Classification (NuSVC), and support vector machine linear [30]. The hyperplane shape was set to radial basis function for SVC and NuSVC. However, NuSVC repeatedly failed to train, and so is omitted from the discussion. The SVM consistently achieves an accuracy of above 80% across all files and only varies by, at most, 1.3%. As shown in Figure 10, the accuracy is consistent when trained with the whole, 1st, and 2nd derivatives of the data. Setting the number of classes to 2 achieves a slightly higher training accuracy of around 86%, compared to approximately 82% for 4 classes for both the 2-class and 4-class model types. The SVMs were also trained using a bagging classifier. A bagging classifier fits multiple classifiers to random subsets of the data and then combines the results of the individual predictions to arrive at a single conclusion [30] using an algorithm known as Pasting [39].

The SVC also achieves an accuracy of above 85% consistently across all files, except SVC normal for 4 classes where the accuracy is close to 82%. The accuracy is consistent when trained with the whole, 1st, and 2nd derivatives of the data, as depicted in Figure 11. Setting the number of classes to 2 achieves a slightly higher training accuracy of around 86%, compared to approximately 82% for 4 classes for both the 2-class and 4-class model types.

As promising as these training accuracy scores were, they were misleading due to the size of the wires. Most of the space in the data recorded was the background, and the wires took up a relatively small part of it. When the training accuracy appeared to be high, this more likely meant that the background was correctly identified and did not necessarily mean that the enamel, if any, was found. This is explained in further detail with the help of three examples shown in Figure 12. The first image in Figure 12a has high accuracy as the four wires are correctly identified in the picture, whereas in Figure 12b wires have been identified but are included in the same class as the misidentified background, giving it a false high accuracy. The majority of the image is background, and wires have been misidentified as background in Figure 12c, again giving it a false high accuracy. For this case, a better metric for training accuracy has been adopted based on the proportion of the enamel correctly identified. For two classes, the locations of the non-zero pixels are compared against the non-zero pixels in the training mask. The new accuracy metric is the proportion of the locations which matches the training mask. For 4 classes, the location of the pixels with a value of 1 is compared to the training mask.

The average enamel classification accuracy for each trained model type for the two-class model is presented in Table 1, and for the four-class model is presented in Table 2. Another means of assessing the models is how long it takes to evaluate the data. The trained model should be able to process live data as fast as it can. The time is measured as how long it takes for the function to load the data file, clip to the cooling period, convert it to a feature set, evaluate it, and return the results. Each model was tested 10 times for each dataset to get a representative spread of data. The average time for a model to evaluate data is presented in Table 3. The best model for the two classes is potentially the GMM. When given the 2nd derivative of the cooling period, it achieves a training accuracy of 85% and an enamel classification accuracy of 100% for the 2-class model. It also has the fastest evaluation time of 1.05 s. For the 4-class model, the SVC with bagging classifier achieved training and enamel classification accuracies of more than 82% when the 2nd derivative of the cooling period was given. However, the evaluation time of 134 s makes it difficult to use for live systems. The two-class and four-class enamel accuracy of SVM linear decreases drastically when the derivative is given to it instead of the cooling period, suggesting that there is a high degree of overlap between the classes and that a linear kernel cannot reliably separate them.

## 5. Conclusions and Future Work

Within electric machine manufacturing for aerospace applications, the individual processes of stripping and joining for making terminations increase the handling effort, production times, and costs. However, they are inevitable in wires that are required to operate effectively at higher temperatures for aerospace applications. A review of the literature and discussions within the industry revealed a lack of any significant research in the area of inspection during the manufacturing of terminations. This work attempts to address this gap by utilising infrared thermal imaging and machine learning techniques for inspection of the enamel removal process on Litz wire, commonly used for aerospace and automotive applications. Infrared thermal imaging was utilised to inspect bundles of Litz wires containing those with and without enamel, and temperature profiles of the wires were recorded. After that, the feasibility of several classifier models (an SVM, a k-means, and a GMM) for automated inspection of enamel removal was investigated. After comparing the performance of the classifier models in terms of classification accuracy, the Gaussian Mixture Model with expectation maximisation was found to be the best with a training accuracy of 85% and enamel classification accuracy of 100%, and the fastest evaluation time of 1.05 s. A future study would involve building a demonstrator prototype by utilising the current finding and applying the Gaussian Mixture Model. The development of a graphical interface for displaying the input and results will also be considered. The demonstrator could be tested on the shop floor of an industrial setting and could serve as part of a live system.

## Figures and Tables

**Figure 1 sensors-23-03977-f001:**
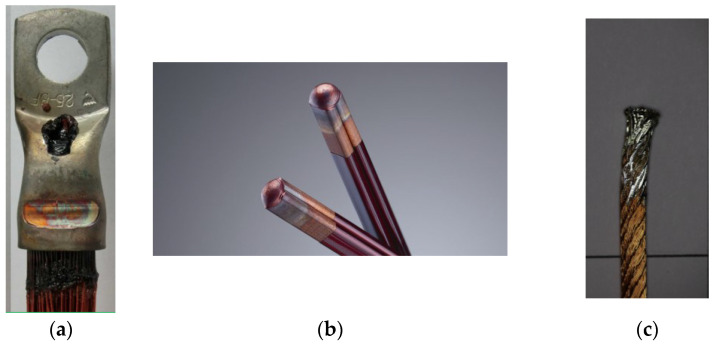
Images showing a few examples of terminations made from various technologies (**a**) Hot crimped termination (**b**) A laser-welded joint (**c**) A dip-soldered termination.

**Figure 2 sensors-23-03977-f002:**
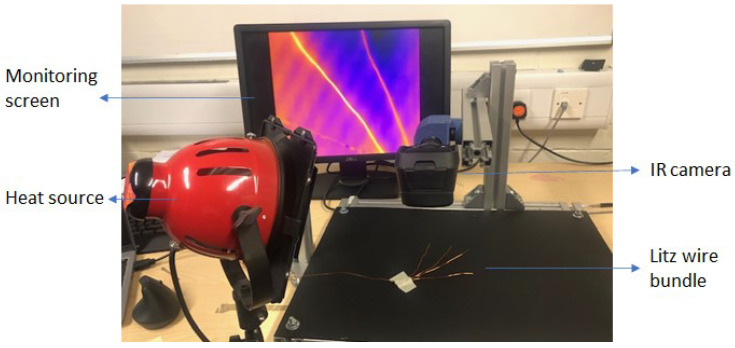
A picture of the setup used for experiments using the IR camera.

**Figure 3 sensors-23-03977-f003:**
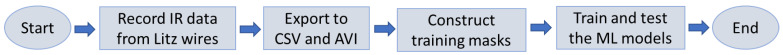
The flowchart depicts the process followed for recording and training models. The IR radiation from the wires was recorded during the heating and cooling cycles. The data obtained were used to construct training masks and train the machine learning models for enamel classification.

**Figure 4 sensors-23-03977-f004:**
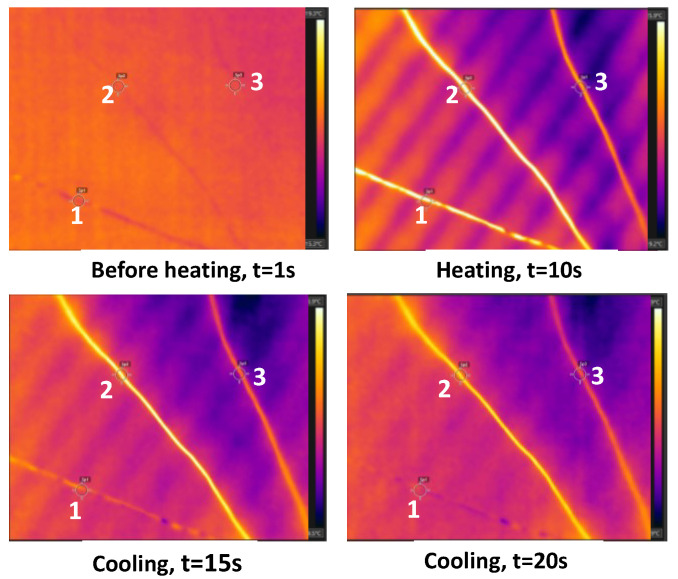
The infrared images of the three Litz wires (labelled 1, 2 and 3) from a bundle during heating and cooling. The wires labelled as 1 had their enamel completely removed, wire 2 had a coating of enamel on the surface, and wire 3 had its enamel coating partially removed.

**Figure 5 sensors-23-03977-f005:**
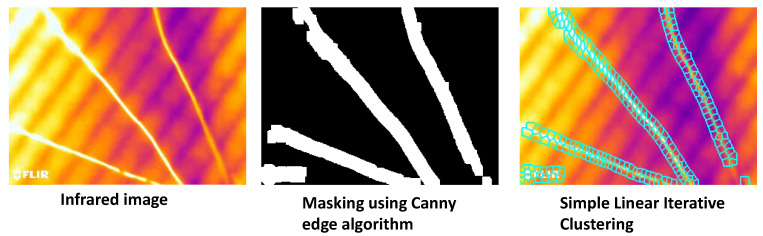
(**left**) An infrared image from heated wires, (**centre**) masked image using Canny edge detection algorithm, (**right**) image obtained using simple linear iterative clustering.

**Figure 6 sensors-23-03977-f006:**
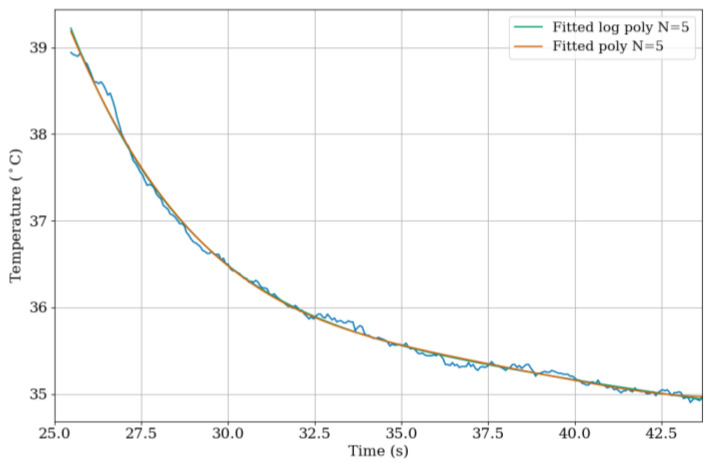
A plot showing TSR fitting to recorded data using 5th-order polynomial and log polynomial. The blue line denotes the raw data.

**Figure 7 sensors-23-03977-f007:**
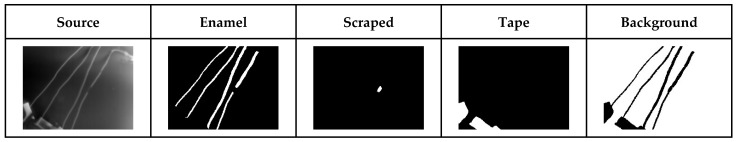
Example of material masks constructed and source image.

**Figure 8 sensors-23-03977-f008:**
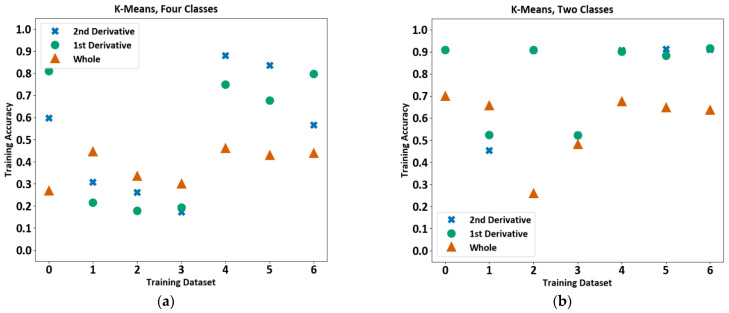
K-means training accuracy for each file using both the cooling period and derivatives of the cooling period. (**a**) The plot is showing results for four classes (copper, enamel, background, and tape), and (**b**) the plot is showing results for two classes (copper and enamel).

**Figure 9 sensors-23-03977-f009:**
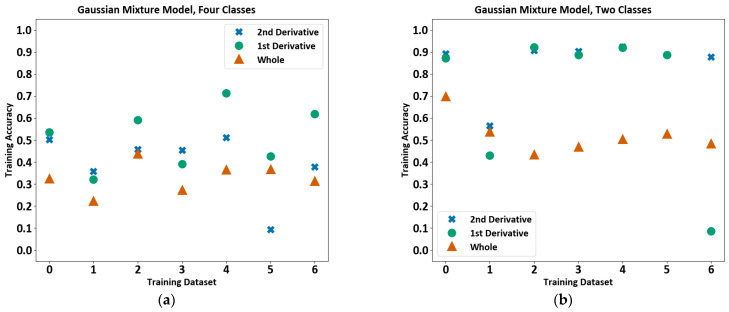
Gaussian Mixture Model training accuracy for each file using both the cooling period and derivatives of the cooling period. (**a**) The plot shows results for two classes, and (**b**) the plot shows results for four classes.

**Figure 10 sensors-23-03977-f010:**
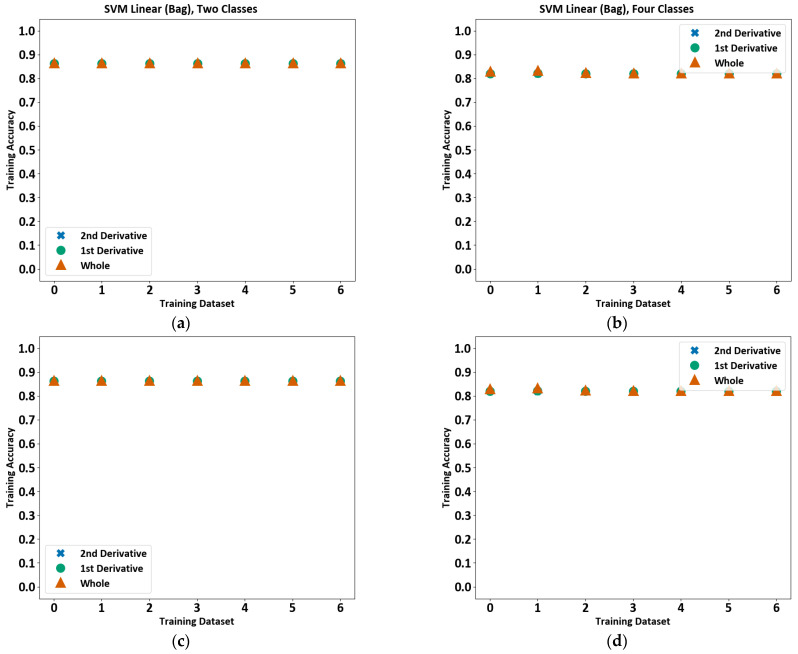
Support vector machine training accuracy for each file. The training accuracy for SVM linear using a bagging classifier for two-class model (**a**), SVM linear using a bagging classifier for four-class model (**b**), SVM linear for two-class model (**c**), and SVM linear for four-class model (**d**) are shown.

**Figure 11 sensors-23-03977-f011:**
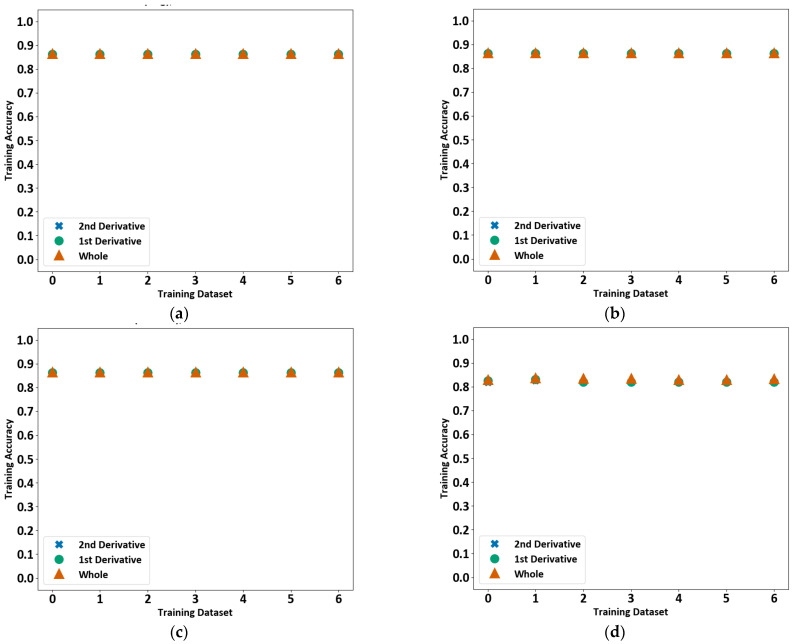
Support vector classification training accuracy for each file. The training accuracy for SVC using a bagging classifier for two class (**a**), SVC using a bagging classifier for four class (**b**), SVC normal for two class (**c**), and SVC normal for four class (**d**) are shown.

**Figure 12 sensors-23-03977-f012:**
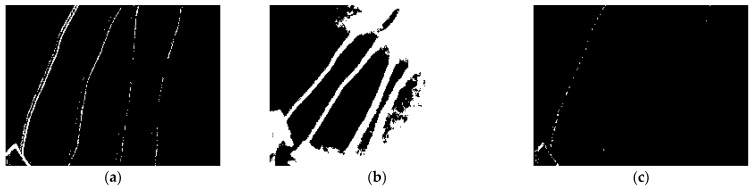
Examples of evaluated results using the trained models. (**a**) the four wires are correctly identified in the picture, giving it a high accuracy (**b**) wires have been identified but are included in the same class as the misidentified background, giving it a false high accuracy (**c**) the wires have been misidentified as background but it gives a false high accuracy as the majority of the image is background.

**Table 1 sensors-23-03977-t001:** Average training and enamel classification accuracy for each trained model type for two classes.

Two Classes						
Class	Whole		1st Derivative		2nd Derivative	
	Average Training Accuracy	Average Enamel Accuracy	Average Training Accuracy	Average Enamel Accuracy	Average Training Accuracy	Average Enamel Accuracy
GMM	0.52	0.32	0.71	0.86	0.85	1.0
K-Means	0.58	0.43	0.79	0.23	0.79	0.55
SVM Linear (Bag)	0.86	0.22	0.86	0.52	0.86	0.37
SVM Linear (Normal)	0.86	0.72	0.86	0.42	0.86	0.65
SVC (Bag)	0.86	0.36	0.86	0.44	0.86	0.51
SVC (Normal)	0.86	0.68	0.86	0.31	0.86	0.62

**Table 2 sensors-23-03977-t002:** Average training and enamel classification accuracy for each trained model type for four classes.

Four Classes						
Class	Whole		1st Derivative		2nd Derivative	
	Average Training Accuracy	Average Enamel Accuracy	Average Training Accuracy	Average Enamel Accuracy	Average Training Accuracy	Average Enamel Accuracy
GMM	0.41	0.26	0.51	0.41	0.39	0.24
K-Means	0.47	0.22	0.52	0.24	0.52	0.64
SVM Linear (Bag)	0.82	0.43	0.82	0.40	0.82	0.64
SVM Linear (Normal)	0.82	0.42	0.82	0.40	0.82	0.63
SVC (Bag)	0.74	0.40	0.59	0.19	0.82	0.83
SVC (Normal)	0.82	0.25	0.82	0.17	0.82	0.47

**Table 3 sensors-23-03977-t003:** Average time for a model to evaluate data. This average includes processing whole, 1st derivative, and 2nd derivative datasets.

Average Model Evaluation Time (s)	
K-Means	4.27
SVM Linear	3.71
SVM SVC	134.47
OpenCV GMM	1.05

## Data Availability

The underlying data can be accessed at https://doi.org/10.15131/shef.data.22596742.

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
