# Peer review of "Inspection of Enamel Removal Using Infrared Thermal Imaging and Machine Learning Techniques"

_sensors, 2023, doi:10.3390/s23083977_

Round 1

Reviewer 1 Report

I have some remarks for the authors:

- It is necessary to explain abbreviations in the text, such as ML, ANN, and CNN.

- The introductory description of sections 2 and 3 is missing.

- The reference to the figure should not be in the form of in "Figure 1“ but in "Fig. 1"., similarly for other images. The phrase Figure is used at the beginning of a sentence.

- Figure 3, wire 3 had a darker color all the time. What caused it?

-  In the text “of 5th, 6th, and 7th“, th should be a superscript everywhere.

- The reference to Figure 6 is missing in the text. 

-   Figure 7 is divided into two pages.

Reviewer 2 Report

Your manuscript is well designed and all the parts are well discussed. 

Author Response

We thank the reviewer for their feedback. There are no comments to address.

Reviewer 3 Report

Please address all comments in the attached file. Modify the abstract and introduction according the comments and provide more new papers.

Author Response

Please see the attachment and Appendix A.

Reviewer 4 Report

This paper developed a metho for inspection of enamel removal using infrared thermal imaging and machine learning techniques. The topic is interesting and valuable. The results showed that the best model for enamel classification accuracy was the Gaussian Mixture Model with Expectation Maximisation. Overall, the paper is well written. I have some minor comments here.

1. the test proposed in this paper is simple. Did you consider the influence of distance, angle and enviroment temperature? 

2. Can the proposed method work under moving situation? (the object is moving)

3. The training set should  contain the samples collected at different situations.

4. Did you consider using neural network to classify the results?
